# Candidate Biomarkers for the Detection of Serious Infections in Children: A Prospective Clinical Study

**DOI:** 10.3390/children9050682

**Published:** 2022-05-07

**Authors:** Maria Chiara Pellegrin, Arturo Penco, Leonardo Amadio, Samuele Naviglio, Luigina De Leo, Oriano Radillo, Gianni Biolo, Nicola Fiotti, Filippo Mearelli, Marco Rabusin, Egidio Barbi, Lorenzo Monasta

**Affiliations:** 1Institute for Maternal and Child Health—IRCCS Burlo Garofolo, 34137 Trieste, Italy; mariachiara.pellegrin@gmail.com (M.C.P.); leonardo.amadio@gmail.com (L.A.); samuele.naviglio@burlo.trieste.it (S.N.); luigina.deleo@burlo.trieste.it (L.D.L.); oriano.radillo@burlo.trieste.it (O.R.); marco.rabusin@burlo.trieste.it (M.R.); egidio.barbi@burlo.trieste.it (E.B.); lorenzo.monasta@burlo.trieste.it (L.M.); 2Department of Medical, Surgical and Health Sciences, University of Trieste, 34127 Trieste, Italy; 3Unit of Internal Medicine, Department of Medical Surgical and Health Sciences, University of Trieste, 34127 Trieste, Italy; gianni.biolo@asugi.sanita.fvg.it (G.B.); fiotti@units.it (N.F.); filippo.mearelli@asugi.sanita.fvg.it (F.M.)

**Keywords:** biomarkers, serious bacterial infection, pediatric, diagnosis, children

## Abstract

Serious bacterial infections (SBI) in children are associated with considerable morbidity and mortality, and their early identification remains challenging. The role of laboratory tests in this setting is still debated, and new biomarkers are needed. This prospective, observational, single-center study aims to evaluate the diagnostic role of blood biomarkers in detecting SBI in children presenting with signs of systemic inflammatory response syndrome (SIRS). A panel of biomarkers was performed, including C-reactive protein (CRP), procalcitonin (PCT), white blood cell count (WBC), absolute neutrophil count (ANC), interleukin (IL)-6, IL-8, IL-10, human terminal complement complex (C5b-9), Plasmalemma-Vesicle-associated protein 1 (PV-1), Intercellular Adhesion Molecule-1 (ICAM-1), and Phospholipase A2 (PLA2). Among 103 patients (median age 2.9 years, 60% males), 39 had a diagnosis of SBI (38%). Significant predictors of SBI were CRP (*p* = 0.001) and ICAM-1 (*p* = 0.043). WBC (*p* = 0.035), ANC (*p* = 0.012) and ANC/WBC ratio (*p* = 0.015) were also significantly associated with SBI in children without pre-existing neutropenia. ROC curves, however, revealed suboptimal performance for all variables. Nevertheless, a model that combined CRP and ANC/WBC ratio had more in-depth diagnostic accuracy than either of the two variables. Overall, this study confirms the limited usefulness of blood biomarkers for the early diagnosis of SBI. WBC, ANC, ANC/WBC ratio, CRP, and ICAM-1 showed the best, albeit moderate, diagnostic accuracy.

## 1. Introduction

Fever is among the most frequent reasons for medical consultation in children. Between 15% and 25% of pediatric consultations in primary care and emergency departments are due to febrile illnesses [1,2,3]. While most of the children will have mild, self-resolving illnesses, a significant proportion of them may harbor a potentially severe infection. In children, serious bacterial infections (SBIs) usually consist of bacteremia, meningitis, pneumonia, urinary tract infection, osteomyelitis, suppurative arthritis, soft tissue infections and bacterial gastroenteritis [4]. SBI rate in children with fever is between 8.5% and 12% [5], and it is even higher in infants younger than three months, with a range of up to 20% [6]. Mortality and morbidity due to SBIs are still elevated. Therefore, it is essential not to underestimate children with more severe diseases [7,8,9]. The diagnosis of SBI is primarily based on clinical presentation. Although several clinical “red flags” can help doctors identify children with SBI, their performance is known to be less than optimal [10,11].

The role of laboratory markers in the early recognition of SBI is still debated. Several mediators (either pro-inflammatory or anti-inflammatory) are released in the host’s first response to infectious insults. Some of these biomarkers, such as C-reactive protein (CRP) and procalcitonin (PCT), have been proved to aid in diagnosing SBIs in clinical practice, yet their sensitivity and specificity are not optimal [12,13,14]. Despite its limits, such as its relatively late appearance and persistence for a more extended period [9], CRP is probably the most widely used diagnostic and prognostic biomarker [9]. PCT may have a higher predictive value for detecting invasive bacterial infections (defined as bacteremia or meningitis) and may be more accurate than CRP in the very first hours of fever [9,14,15,16,17,18]. Other biomarkers involved in inflammation, including endothelial and complement activation molecules, coagulation, organ dysfunction and apoptosis, have been proposed as a possible tool for SBI diagnosis, monitoring, and prognosis [18,19]. However, their role is still unclear. The present study aims to evaluate the diagnostic performance of a panel of biomarkers for SBI diagnosis in pediatric patients.

## 2. Materials and Methods

### 2.1. Patients and Data Collection

This monocentric prospective observational study was performed between May 2016 and July 2017 at the Institute for Maternal and Child Health IRCCS Burlo Garofolo in Trieste, Italy, according to the ethical guidelines of the 1975 Declaration of Helsinki (2008 revision). The Institutional Ethics Committee (n. 56/2013) approved it and written informed consent was signed by parents or legal guardians.

The investigated variables were the acute phase reactants and cytokines levels in relation to the diagnosis of severe bacterial infection. Pediatric patients (<18 years of age) consecutively admitted to the Emergency Department or the Hematology–Oncology ward of our Institute were considered eligible if they met the following inclusion criteria: children having at least two out of the four clinical criteria for systemic inflammatory response syndrome (SIRS, Table 1) [20] plus a clinical indication to take a blood sample. Within 24 h of admission, blood serum samples were obtained from each patient. Blood cultures were acquired before starting antibiotic therapy. Clinical and laboratory data were collected prospectively without interfering with the clinical practice. The clinical diagnosis made by the attending physician at the patient’s discharge was recorded, and only patients with a final diagnosis (clinical or microbiological) of an infectious disease were included in the analysis. The primary outcome was the presence of an SBI, as indicated in the clinical diagnosis at discharge. For the purposes of this study, SBIs included bacteremia, meningitis (defined as isolation of a bacterial pathogen in the cerebrospinal fluid), pneumonia, urinary tract infection (defined as significant bacteriuria with growth of ≥100,000 colony forming units (CFU) of a single uropathogen, or ≥100,000 CFU/mL of one uropathogen and <50,000 CFU/mL of a second uropathogen associated with significant pyuria), osteomyelitis, suppurative arthritis, soft tissue infection, and bacterial gastroenteritis (defined by the presence of compatible clinical findings and a positive stool culture for a significant intestinal bacterial pathogen) [4]. Exclusion criteria were refusal to provide written informed consent and the lack of a clinical indication for blood sampling. The sample size was estimated on the basis of the previous literature, with numbers of patients in similar studies ranging from 50 to 347 [12,15,17,18] and considering the limitation of time for the study.

While CRP, PCT and white blood cell count are the usual tests, their predictive power is limited overall. We therefore decided to investigate a panel of biomarkers, considering those most recently proposed in the literature on adults and children.

The study workflow is shown in Figure 1.

### 2.2. Biomarker Determination

A blood sample for biomarkers was taken at enrolment, and serum aliquots were stored at −80 °C. Determinations of biomarkers were carried out according to manufacturers’ instructions and by a single laboratory technician (LA), who was not aware of the patient’s clinical presentation. The nine candidate biomarkers included were: C-reactive protein (CRP), procalcitonin (PCT), interleukin-6 (IL-6), interleukin-8 (IL-8), interleukin-10 (IL-10), secreted type IIA phospholipase A2 (PLA2), human terminal complement complex (C5b-9), intercellular adhesion molecule 1 (ICAM-1) and plasmalemma vesicle-associated protein 1 (PV-1). CRP was analyzed by the Beckman Coulter AU480 System CRP Latex reagent (Brea, CA, USA), with a range of 2–1600 mg/L. PCT was tested using an electrochemiluminescence method (Elecsys Brahms PCT, Roche diagnostic, Mannheim, Germany), measuring 0.02–100 ng/mL. Soluble PLA2 was assessed with a PLA2 ELISA kit from Cusabio Biotech (Wuhan, China). The detection range was 0.45–30 ng/mL. ICAM-1 was analyzed with ICAM-1 (CD54) SimpleStep ELISA kit from ABCAM (Cambridge, UK). The minimal detectable dose was 1.6 pg/mL. C5b-9 was assessed with a TCC C5b-9 ELISA kit from BlueGene Biotech (Shanghai, China). The detection limit was 1.0 ng/mL. IL-6, IL-8 and IL-10 were evaluated, respectively, with Human IL-6 Platinum, Human IL-8 Coated, and Human IL-10 Platinum ELISA kit from Invitrogen by Thermo Fisher Scientific (Waltham, MA, USA); the detection limits were 0.92 pg/mL, 2.0 pg/mL and 1.0 pg/mL, respectively. PV-1 was determined by ELISA assay (Immunosorbent Assay Kit, Biomatik, Cambridge, ON, Canada). The detection range was 0.15–10 ng/mL. All measured results were considered as continuous variables, except for PLA2, coded as a dichotomous variable (values above or below 4.5 ng/mL).

### 2.3. Statistical Analysis

Continuous variables were reported as medians and interquartile ranges (IQR), and categorical variables as the number of observations and percentages. Comparisons of continuous variables among categories were performed with the Mann–Whitney U test. Associations between categorical variables were assessed with Fisher’s two-tailed exact test. Bivariate and multivariate logistic regression models were created to identify significant associations between biomarkers and the diagnosis of SBI, and to calculate the odds ratios (OR) of unitary increases of each variable with the primary outcome. Receiver operator characteristic (ROC) curves detected the performance of selected biomarkers in identifying patients with SBI. Since immunodepression (e.g., cancer chemotherapy resulting in neutropenia) could negatively affect the release of several of the tested biomarkers, statistical analyses were performed both in the whole study sample and in non-immunocompromised patients (i.e., in patients not being treated for an underlying malignancy). Statistical analyses were conducted with Stata/IC 14.2 for Windows (StataCorp LLC, College Station, TX, USA).

## 3. Results

One hundred and three patients were enrolled (males = 62.6%) during the study period; 86 were in the emergency department, and 17 were in the hematology–oncology ward. Thirteen (13%) were infants <2 months of age, 17 were immunocompromised subjects (16 of them had a central venous line). A diagnosis of SBI was made in 39/103 patients (38%) in the whole study population (33/86 in the non-immunocompromised subgroup). Blood cultures were performed in 62 patients (60%), of which 8 (13%) were positive, and all were in the group of SBI. Detailed characteristics of the patients are summarized in Table 2. The final diagnoses of patients with SBIs are reported in Table 3.

No deaths occurred during the study period. Seventeen patients were affected by iatrogenic immune deficiency due to cancer treatment. The biomarkers levels, determined at enrolment blood sampling, are reported in Table 4 and Table 5 for the whole study population and the non-immunocompromised subgroup, respectively. Not all biomarkers were available for all patients due to the scarcity of sampled blood in some patients and the complex blood collection.

### 3.1. Association between Single Biomarkers and SBI

Variables significantly associated with SBI at patient’s admission at bivariate logistic regression were CRP (OR 1.13 [95% CI 1.05–1.21]; *p* = 0.001) and ICAM-1 (OR 1.006 [95% CI 1.0001–1.0118]; *p* = 0.043). An association was also found of an increased PCT in patients with SBI (*p* = 0.056), but it did not reach statistical significance.

### 3.2. Biomarkers in SBI in Non-Immunocompromised Patients

In the group of non-immunocompromised patients, CRP (OR 1.156 [95% CI 1.065–1.256]; *p* = 0.001) and ICAM-1 (OR 1.0069 [95% CI 1.0004–1.0134]; *p* = 0.037) were also significantly higher in patients with SBIs; in addition, White Blood Cells (WBC) count (OR 1.077 [95% CI 1.005–1.155]; *p* = 0.035; OR calculated as per 10^3^ cells/mcL), Absolute Neutrophil Count (ANC) (OR 1.096 [1.021–1.171]; *p* = 0.012; OR calculated as per 10^3^ cells/mcL) and ANC/WBC ratio (OR 1.032 [1.007–1.057]; *p* = 0.015) were significantly higher in the group with SBI than non-SBI. An association between increased PCT and a diagnosis of SBIs was also found in this patients group, but it did not reach significance (*p* = 0.064). Multivariate regression analysis could not be conducted due to the lack of values for several biomarkers.

The diagnostic performance of selected biomarkers in identifying children with SBIs was evaluated using ROC curves (Table 6). Most ROC curves had an acceptable (i.e., between 0.7 and 0.8) value for the area under the curve (AUC) analysis, thus indicating a suboptimal execution for clinical purposes.

### 3.3. Combination of Biomarkers and Diagnosis of SBI

We further explored the possibility that a combination of biomarkers could accurately identify patients with SBI. No combination achieved better performance than single biomarkers when considering the whole study population. However, when the analysis was limited to non-immunocompromised patients, the combination of CRP and ANC/WBC ratio improved AUC. The following logistic regression equation expressed the best performing combination between these two variables:y = 1/(1 + EXP(−(−3.337734 + 2.922786 × (ANC/WBC) + 0.14339 × CRP)))

A value of y equal to or less than 0.22 resulted in 100% sensitivity in identifying children with SBI, with a 47% specificity.

## 4. Discussion

In this study, no single biomarker, out of those considered, yielded a satisfactory predictive power for detecting SBI in children, with better performance being provided by those already used in clinical practice, like WBC, CRP and PCT.

The utility of biomarkers for the early diagnosis of SBI in children is debated. Several mediators and different molecules have been proposed in preclinical or clinical studies, but no gold standard has been identified for pediatric patients, and studies are still relatively limited [8,21,22,23,24,25]. It seems unlikely that one single molecule could be used as an optimal biomarker for diagnosing an SBI [21,26,27]. To date, no new biomarkers (including cytokine/chemokine, cell surface receptors, coagulation and complement molecules, markers of vasodilation or organ disfunction) are more efficient than those already used in clinical practice (CRP, PCT and WBC, ANC/WBC and the ratio between immature and total neutrophils numbers) [28,29,30,31,32]. A possible pitfall when considering candidate biomarkers is their increase and decrease timing, showing a wide range of variations. IL-6, IL-8 and IL-10 peak as soon as sepsis is suspected, while PCT and CRP react later, with concentration peaks at 8–16 and 16–24 h, respectively [33]. Moreover, it is often difficult to determine the exact timing of the measurement in real-life conditions. While many biomarker studies in febrile children have led to a diagnosis of sepsis, SBI is far more frequent. For this reason, identifying an effective biomarker could be extremely useful in directing patient care, thus avoiding unnecessary diagnostic tests, limiting health care costs, antibiotic therapies, and misdiagnosis of severe infection.

In this study, CRP only showed optimal diagnostic accuracy for the diagnosis of SBI, while PCT was only marginally helpful. Previous reports have revealed some effectiveness of PCT as an infection marker and for antibiotic stewardship in adults [34,35]. A recent meta-analysis evaluating the diagnostic accuracy of PCT as a biomarker of SBI in feverish children concluded that PCT did not have sufficient sensitivity or specificity. However, it had better accuracy than CRP and WBC count in identifying SBI in children, and it could perform better in identifying the most severe infections, such as bacterial meningitis and sepsis [36]. In our study, PCT had only moderate accuracy in diagnosing SBI in children, and it ultimately appeared to perform worse than CRP.

Even though the use of CRP could be limited by its relatively late rise, which can only occur after 18–24 h from symptoms onset, WBC and absolute neutrophil counts may grow more rapidly and are widely employed in the routine diagnostic workup of SBI, but they have less diagnostic capacity than CRP or PCT. One study showed that WBC count did not add additional information over CRP and PCT in multivariate logistic regression analysis [37]. In our study, ANC/WBC ratio, when evaluated in non-leukopenic patients, also had a limited diagnostic performance, as demonstrated by its ROC curve AUC. However, when combined with CRP, it resulted in greater accuracy than either marker taken alone.

The percentage of SBI in our population was quite high. This was possibly due to the prospective design of the study, which included the enrollment of patients with SIRS criteria undergoing blood sampling on the grounds of clinically suspected SBI. On this ground, well-appearing patients would have likely not been included in the sample. This approach will have likely selected a population of patients more at risk of having an SBI, but these data do not seem to affect the results of the study.

The performance of candidate biomarkers (IL-6, IL-8, IL-10, C5b-9, PV-1, PLA2) in detecting SBI in children was poor. The best was ICAM-1. ICAM-1 is expressed by the endothelial cell activation during induction and progression of the systemic response. This activation may lead to changes in leukocyte trafficking, vascular permeability, inflammation and microcirculatory flow, which may contribute to organ damage [38,39]. ICAM-1 is part of the cell surface immunoglobulin superfamily of adhesion receptors, which contributes to lymphocyte-mediated adhesion, cytotoxic T-cell activity, antigen presentation, and a ligand for macrophage-associated complex (MAC-1) [38]. Previous studies suggest a role in the early diagnosis of sepsis in newborns and infants, showing that it may be associated also with its severity degree [40,41]. On the other hand, a study on adult patients admitted to the emergency department with sepsis showed an association between levels of biomarkers of endothelial activation and sepsis severity, organ dysfunction and mortality [38]. Our research also evaluated PV-1, an endothelial protein that has a crucial role in endothelial permeability and leukocyte migration in normal and pathologic conditions [42,43,44], but it did not find an association with SBIs.

The strengths of this study include the prospective design and the evaluation of some biomarkers that had never been tested before in pediatric patients with SIRS. Furthermore, only a few studies have evaluated biomarkers for the diagnosis of SBIs, while the majority are focused only on the occurrence or the severity of sepsis. However, some limitations need to be pointed out. The study was conducted in two different real-life settings, the emergency department and the oncology ward. For this reason, we performed a separate analysis on non-immunocompromised patients (i.e., most patients presenting to the emergency department) to avoid possible biases due to immune suppression.

A further significant limitation was the small number of patients enrolled (103), and with an SBI (39). Another major study limitation was the impossibility of testing all biomarkers in all patients because of the scarcity of sampled serum, primarily due to difficulties in blood sampling in children in the ED; this affected the statistical analysis in evaluating the performance of some biomarkers and prevented a multivariate analysis from being performed. This concern was particularly true for ICAM-1, one of the most promising candidate biomarkers according to our analysis, but with the lowest numbers. Overall, the small sample size and the impossibility of testing all biomarkers in all patients limits the possibility of drawing definitive conclusions on the role of the tested biomarkers, and our results should therefore be considered exploratory and hypothesis-generating for further studies.

A small age difference was found between the group of patients with and without a diagnosis of SBI, with the former being younger than the latter. When excluding immunocompromised patients, this difference reversed. The underlying causes of this difference are unclear, and can be attributed to a relative lack of numbers, leading to selection bias, as SBIs are recognized to occur more frequently in younger children. Another limitation of our study is the missing value of time elapsed between symptom onset and presentation to the emergency department. This could affect the value of some biomarkers, as their increase at different moments after the onset of infection may have a different clinical significance. We were also not able to perform repeated assays at different time points for each patient, because several children would not have had a clinical indication for repeated blood sampling. Evaluation of the evolution of biomarkers at different timings during the course of infection, in fact, could be interesting, as each biomarker may have special kinetics that cannot be addressed by a single measurement and various studies in children have described dynamic changes in the early phases of sepsis, with associated changes in mortality and morbidity [45]. While this could definitely represent a starting point for further studies, the main purpose of the study was to evaluate the value of these biomarkers to aid identification of serious infections upon clinical presentation (i.e., in the emergency department) rather than during the course of disease.

We did not perform weight or age adjustment for the evaluated biomarkers. While weight adjustment is seldom used for biomarkers in pediatrics, age adjustment may be important for some values. The limited size of the study population and the missing values for some biomarkers, did not allow us to further stratify patients according to age in order to evaluate the diagnostic performance of each biomarker. Finally, the examined study population was heterogeneous regarding the focus of bacterial infection. In fact, it cannot be excluded that a specific biomarker may have a greater diagnostic value for a specific type of infection, while not displaying a good performance overall.

## 5. Conclusions

This study confirms the limited utility of biomarkers for diagnosing SBIs in children. Among the candidate biomarkers, none reached statistical significance, except for ICAM-1, but this result could be affected by the low number of samples in the population and should be confirmed in more extensive studies; therefore, these conclusions should be considered exploratory. Among the biomarkers already used in clinical practice, our study confirms their moderate predictive value, with the best performance provided by the combination of WBC/ANC ratio and CRP, two variables already normally available in clinical practice. These outcomes highlight the fact that, at present, clinical judgement remains the mainstay of SBI diagnosis in children.

## Figures and Tables

**Figure 1 children-09-00682-f001:**
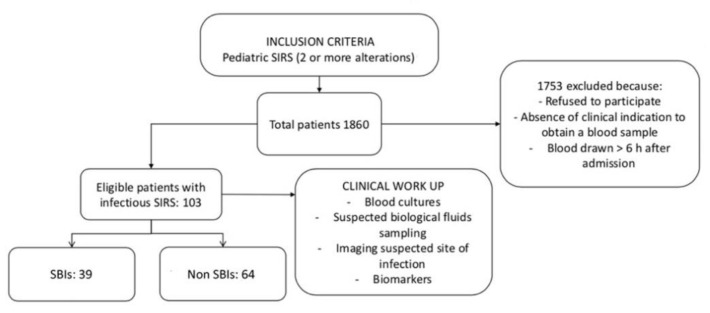
Study design flow chart.

**Table 1 children-09-00682-t001:** SIRS criteria according to age (modified from reference [20]).

Age	Heart Rate (bpm)	Respiratory Rate (apm)	Body Temperature (°C)	White Blood Cells (cells/mcL)
<7 days	<100/>180	>60	<36/>38	>34,000
7–30 days	<100/>180	>50	<36/>38	<5000/>19,500
30 days–2 years	<90/>180	>35	<36/>38	<5000/>17,500
2–5 years	>140	>30	<36/>38	>6000/>15,500
5–12 years	>130	>20	<36/>38	<4500/>13,500
12–18 years	>100	>20	<36/>38	>4500/>11,000

**Table 2 children-09-00682-t002:** Demographic and clinical characteristics of the population.

	All	SBI	Non SBI	*p*-Value
Number	103	39	64	
Age (years)	2.9 (0.7–7.8)	2.6 (0.4–9.9)	3.7 (1.6–12.4)	0.029
Gender (male)	62 (60)	23 (59)	39 (61)	n.s. ^1^
Comorbidity	29 (28)	15 (39)	14 (22)	n.s.
Temperature (°C)	38.4 (38.0–39.0)	38.5 (38-0–38.9)	38.4 (38.0–39.0)	n.s.
HR (bpm)	140 (120–160)	137 (120–152)	140 (120–165)	n.s.
RR (apm)	34 (25–43)	36 (25–45)	34 (25–40)	n.s.
SpO_2_	98 (97–99)	98 (96–99)	98 (97–99)	n.s.

^1^ n.s. = not significant, defined as *p*-value < 0.05. Data reported as median (interquartile range) or absolute number (percentage).

**Table 3 children-09-00682-t003:** Final diagnoses of SBIs in patients.

**Diagnosis**	**Number**
Bacteremia	7
Meningitis	1
Urinary tract infection	4
Pneumonia	11
Complicated pneumonia	6
Osteomyelitis	6
Soft tissue infection	2
Bacterial gastroenteritis	4
Total	39

**Table 4 children-09-00682-t004:** Biomarker levels in the studied population.

	All Patients	SBI	Non SBI	*p*-Value	N of Patients for Whom Testing Was Available
*n*	103	39	64		
CRP (mg/L)	30 (11–83)	66 (29–153)	16 (8–53)	0.001	103
PCT (ng/mL)	0.3 (0.1–1.0)	1.0 (0.2–2.2)	0.2 (0.1–0.5)	0.056	80
IL-6 (pg/mL)	18.6 (7.6–42.9)	27.4 (14.4–78.9)	13.1 (5.2–25.8)	0.090	99
IL-8 (pg/mL)	15.6 (10.1–27.8)	15.8 (10.8–24.3)	14.3 (9.9–29.4)	0.305	93
IL-10 (pg/mL)	6.0 (5.0–22.2)	6.0 (5.0–19.7)	6.7 (5.0–24.8)	0.905	79
C5b-9 (μg/mL)	0.6 (0.4–0.7)	0.6 (0.4–0.8)	0.5 (0.4–0.7)	0.153	69
PV1 (ng/mL)	1.8 (0.8–3.2)	1.8 (0.8–3.0)	1.7 (0.7–3.3)	0.644	58
ICAM-1 (ng/mL)	361.8 (291.1–448.8)	486.8 (260.6–636.4)	349.7 (295.0–419.9)	0.043	40
PLA2 (>4.5 ng/mL)	27 (26)	10 (26)	17 (27)	0.918	103

Data reported as median (interquartile range) or absolute number (percentage). *p* Mann–Whitney *U* or Fisher’s exact test for the comparison of SBI and non-SBI. See text for biomarkers abbreviations.

**Table 5 children-09-00682-t005:** Biomarker levels in non-immunocompromised patients.

	All Patients	SBI	Non SBI	*p*-Value	N of Patients for Whom Testing Was Available
*n*	86	33	53		
Age (years)	2.5 (0.3–5.6)	3.3 (1.1–10.5)	2.3 (0.3–4.7)	0.007	
Gender (males)	55 (64)	22 (67)	33 (62)	0.141	
WBC (10^3^/mcL)	11.97 (9.41–18.01)	15.37 (10.39–19.88)	10.91 (9.41–15.20)	0.035	86
ANC (10^3^/mcL)	7.95 (4.80–13.40)	10.10 (6.50–15.40)	7.10 (4.10–9.80)	0.012	86
ANC/WBC (%)	51.9 (20.6–67.3)	64.2 (0.19–0.76)	40.7 (20.6–80.5)	0.015	86
CRP (mg/L)	31 (10–87)	82 (41–153)	15 (7–52)	0.001	86
PCT (ng/mL)	0.3 (0.1–1.1)	1.1 (0.3–5.2)	0.2 (0.1–0.6)	0.064	63
IL-6 (pg/mL)	17.6 (6.8–29.4)	25.8 (11.3–52.1)	11.2 (4.9–25.8)	0.110	82
IL-8 (pg/mL)	12.8 (9.0–18.9)	14.2 (9.9–18.3)	11.9 (8.9–20.0)	0.696	76
IL-10 (pg/mL)	8.0 (5.0–25.4)	6.0 (5.0–19.7)	8.3 (5.0–26.0)	0.809	64
C5b-9 (μg/mL)	0.6 (0.4–0.7)	0.6 (0.4–0.9)	0.5 (0.4–0.7)	0.180	54
PV1 (ng/mL)	2.0 (1.4–3.3)	1.8 (0.9–3.2)	2.0 (1.5–3.8)	0.511	44
ICAM-1 (ng/mL)	394.6 (295.0–506.1)	561.1 (340.4–636.4)	366.7 (295.0–419.9)	0.037	31
PLA2 (>4.5 ng/mL)	25 (29)	9 (27)	16 (30)	0.099	86

Data reported as median (interquartile range) or absolute number (percentage). *p* Mann–Whitney *U* or Fisher’s exact test to compare SBI and non-SBI. See text for biomarkers abbreviations.

**Table 6 children-09-00682-t006:** The area under the ROC curve for selected biomarkers for the diagnosis of SBI.

Biomarker	Area under the ROC Curve, all Patients (95% CI)	Area under the ROC Curve, Non-Immunocompromised Patients only (95% CI)
CRP	0.75 (0.65–0.85)	0.79 (0.69–0.88)
PCT	0.73 (0.61–0.85)	0.74 (0.61–0.88)
ICAM-1	0.64 (0.37–0.90)	0.69 (0.41–0.96)
ANC/WBC ratio	n/a	0.68 (0.56–0.79)
ANC/WBC ratio + CRP model	n/a	0.83 (0.74–0.91)

## Data Availability

The dataset analyzed during the current study is available from the corresponding author on reasonable request.

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
