# Peer review of "Candidate Biomarkers for the Detection of Serious Infections in Children: A Prospective Clinical Study"

_children, 2022, doi:10.3390/children9050682_

Round 1
Reviewer 1 Report
The study presented by Pellegrin et al adds new information about the value of biomarkers in the diagnosis of severe bacterial infections. This is a very relevant and necessary topic to establish a rapid diagnosis of severe infections, related to a better prognosis. The literature to date is extensive and the results have been disappointing, aligned with the previous ones, not contributing with new elements to what is already known.
These are my comments.
Title: reflects the objective, target population and type of study.
Summary: Please, introduce the type of study (prospective, observational, single-centre) and the target population.
Introduction.
The introduction is clear and focused on the research question: “evaluate the diagnostic performance of a biomarker panel for the diagnosis of GBI in pediatric patients”. Secondary objectives are missed. About references, there are more recent papers on the subject, including systematic reviews.
Methodology.
I would call it a prospective observational study.
The variables to be studied are not defined.
The sample size should have been calculated to give validity to this study. It seems quite small.
CRP, PCT along with white blood cell count are the usual tests, when trying to look for other biomarkers it should be explained why these biomarkers have been chosen and not others.
Results
They do not follow a friendly structure, becoming very confusing. The variables they mention are not well-explained in methods.
There are few cases in the different groups, especially those with a higher incidence of infection such as infants with suspected infection, immunocompromised, etc.
Nothing is mentioned about the biomarkers interleukin (IL)-6, IL-8, IL-10, human terminal complement complex (C5b-9), Plasmalemma-vesicle like protein partner 1 (PV-1), and Phospholipase A2 (PLA2). Please could you explain how the final diagnostic equation is constructed?.
Discussion
The discussion does not focus on the results of the study, only superficially mentioning them at some point. It focuses more on giving a list of studies on the usual markers of infection together with a somewhat more extensive list of the role of ICAM-1 and PV-1, it does not mention anything about C5b-9 or PLA2.
The limitations of the study are clear: small sample size.
The study does not provide any new recommendations.
Extensive references, although there are more recent articles on the subject.
Tables and figures.
Table 2 , it is not defined what the values are. (Mean. Median, etc.)
A figure with the ROC curves should be added.
Reviewer 2 Report
The authors describe the evaluation of various biomarkers in a pediatric population with a median age of 2.9 years old. Within 24 hours of hospital admission BLOOD SERUM samples were obtained. The laboratory technician performing the sample analysis was blinded to the clinical results.
The study team did not find added benefit from the biomarker measurements as the markers only provided moderate diagnostic accuracy.
It appears that the enrolled patients already had a higher likelihood of SBI as the rate of identified SBI is with 38 % higher than expected. The authors should comment on this.
Various studies in children have described dynamic changes in the early phases of sepsis with associated changes in mortality and morbidity.
Wong, H.R., Weiss, S.L., Giuliano, J.S. Jr, Wainwright, M.S., Cvijanovich, N.Z., et al. (2014) The Temporal Version of the Pediatric
Sepsis Biomarker Risk Model. PLoS ONE 9(3): e92121. doi:10.1371/journal.pone.0092121
The timing of the sample collection is important and should be highlighted in this concern.
Round 2
Reviewer 1 Report
Thank you for your revised manuscript. Your revisions respond adequately to the suggestions.
This manuscript is a resubmission of an earlier submission. The following is a list of the peer review reports and author responses from that submission.
Round 1
Reviewer 1 Report
The authors describe within their study, the performance of different biomarkers for the detection of serious bacterial infections in children. Even, the subject of the study is quite interesting, several limitations are included and need to be addressed.
Major points:
- The definition of serious bacterial infections is incomplete and needs to be specified. Especially, the urinary tract infections are pointed to unspecific and needs to be separated from bacterial colonization within the definition and the analysis of the study results.
- Within the manuscript demographic data regarding the study population (age, pre-existing illness, weight, height,…) are missing. Only a small part of information is presented in the result section. This point needs to be addressed and more data should be presented. Moreover, information regarding the detected bacteria, as well as the rate of taken blood cultures are not presented. This information in combination with the used antibiotic therapies might be also interesting for the reader to classify the results.
- Only one sample of blood was drawn for measurement of the biomarkers. This point limits the value of the study, as most of the examined biomarkers have a special kinetic, which cannot be addressed by a single measurement. This point needs to be addressed within the discussion.
- The authors present results of biomarkers, which are not measured in all included patients, due to technical reasons. Especially, the most promising biomarker ICAM-1 pointed by the authors, showed the lowest number of samples. This in turn, may falsify the results and overestimate the value of the biomarker.
- The authors found only in 39 of the 103 included children’s positive cultural findings. Maybe the comparison between children with positive cultural findings and them with only the clinical diagnosis shows a better performance of the examined biomarkers.
- The examined study population is very heterogenous regarding the presented focus of the serious bacterial infection. This point might explain the limited results shown within the manuscript and needs also to be addressed within the discussion.
- The conclusion of the authors regarding to ICAM-1 is too optimistic, according to the limited number of analysed samples. The conclusion should be attenuated in a reworking.
Reviewer 2 Report
Thank you for your submission of your well written and interesting manuscript to the MDPI journal diagnostics.
The authors present data of various inflammation related biomarkers from a pediatric patient population meeting two of four SIRS criteria, indications to obtain blood culture samples when presenting to an emergency department or the hematology-oncology ward.
The authors were able to collect samples and data for 103 pediatric patients age 0.7 to 7.8 years. From these prospectively at risk children for serious bacterial infection 38% developed culture positive infections.
While there was significant association of CRP, ICAM-1, WBC, ANC and the ratio of ANC/WBC with SBI, ROC curves highlighted suboptimal performance. These findings are inline with other publications in pediatric sepsis and infections such as the PERSEVERE study and several publications by Hector Wong's group. (Jacobs, L., Berrens, Z., Stenson, E.K. et al. The Pediatric Sepsis Biomarker Risk Model (PERSEVERE) Biomarkers Predict Clinical Deterioration and Mortality in Immunocompromised Children Evaluated for Infection. Sci Rep 9, 424 (2019).)
A one time, weight unadjusted measurement, such as in this manuscript, may not be adding much to the clinical diagnosis of SBI. The brief discussion about the dynamic nature of the cytokine biomarker measurements should also explain why the authors choose not to perform serial measurements.
Do the authors plan to not only use age adjustments but also weight adjustments for the measured biomarkers? Consider adding this to limitations.
The ROC for the combined markers of ANC/ WBC ratio with CRP are quite reasonable at 0.83. You could highlight this in concern that most western hospital laboratories can readily measure these markers.
Formatting Comments
Table 1 may be better presented as a Figure or Graph instead of a table.
Tables should be adjusted to assure that table relevant data stays together. (Table 2, the total is on a separate page from the table information).
Reviewer 3 Report
Authors tried to investigate significant biomarkers for SBI in children with prospective study.
The biomarker study of SBI is important, but the following issues should be considered.
Major issue
The total cases of SBI were only 39, and with these small cases, it is not easy to have significant conclusion in biomarker study.
Minor issues
- They used bivariate logistic regression analysis. Which variables were included in logistic regression analysis? By which criteria? Variables such as fever was not included. Is there any reason?
- There were too many missing values for biomarkers to be checked, so it is difficult to conclude as the authors did. For example, of the total 103 cases, ICAM-1 was checked only in 40 patients, i.e. less than 50%.
Author Response
请参阅附件。

Round 2
Reviewer 1 Report
The authors present a revised version of their manuscript, in which the made suggestions are included. Nevertheless, some points need to be readdressed.
The term CFU could be defined within the introduction.
Age and gender are presented in table 2, table 4 and table 5. One time is sufficient.
Reviewer 3 Report
None